# Food Sales and Adult Weight Status: Results of a Cross-Sectional Study in England

**DOI:** 10.3390/nu14091745

**Published:** 2022-04-22

**Authors:** Stephanie Howard Wilsher, Flo Harrison, Andrew Fearne, Andy Jones

**Affiliations:** 1Norwich Medical School, University of East Anglia, Norwich NR4 7TJ, UK; drfcdharrison@gmail.com; 2Norwich Business School, University of East Anglia, Norwich NR4 7TJ, UK; a.fearne@uea.ac.uk; 3Public Health, Norfolk County Council, Norwich, NR1 2DH, UK; andrew.jones3@norfolk.gov.uk

**Keywords:** adult, obesity, neighbourhood, food sales, environment, deprivation

## Abstract

Ecological studies often use supermarket location as a proxy measure of the food environment. In this study, we used data on sales at a leading mainstream supermarket chain to explore how area-level supermarket use is associated with overweight and obesity in English adults. Sales data were aggregated to local authority level and joined to a national dataset describing self-reported height and weight and fruit and vegetable consumption. Regression models showed a modest association between higher levels of unhealthy food sales relative to health food sales and increased odds of being overweight and obese. Although effect sizes were small, they persisted after adjustment for area-level deprivation. Supermarket sales data provide additional understanding in the study of food environments and their impact on increasing weight status. Future health policies should consider using ‘big data’ combined with other research methods to address the increasing consumption of unhealthy and highly processed foods.

## 1. Introduction

Around 64% of the English population is overweight, and 26% are obese [1]. Obesity greatly increases the risk of many chronic conditions, including diabetes, asthma, and coronary heart disease [2] and is estimated to cost the National Health Service more than GBP 5 billion per year [3]. Environmental factors are thought to be important contributors to recent increases in obesity rates, with ‘obesogenic environments’ providing limited opportunities for physical activity and easy access to foods high in fat and sugar [4]. A comprehensive review noted that highly processed foods, such as cakes, biscuits, and soft drinks, are associated with obesity and health-related outcomes [5].

Research on obesogenic food environments can be broadly grouped into population-level studies utilising large datasets and focusing on the location of certain food outlets [6,7,8,9,10] and smaller-scale behavioural studies that investigate how individuals utilise their neighbourhoods for food purchasing [11,12,13,14]. The former group of studies attempts to explore relationships between food outlet availability and diet and weight status, hypothesising that people living in areas with more unhealthy outlets (typically fast-food restaurants and small grocery/convenience stores) are at increased risk of having diets associated with weight gain. For example, Mazidi and Speakman [10] calculated the density of fast-food and full-service restaurants in counties across the USA using government store location data. They assessed associations between these densities and area obesity prevalence from the Center for Disease Control but found no association after adjusting for socioeconomic status. Findings from similar studies have been mixed; a review undertaken in 2015 found that the results of many such studies were null but that associations between supermarket availability and obesity were typically negative, whereas the associations with fast food outlets were positive [6]. The reasons for these mixed findings are unknown but may be associated with methodological limitations in study design and data utilised.

Population-level studies are attractive, as they are relatively quick and easy and make use of existing data, with potentially large sample sizes. Furthermore, they can cover large geographical areas with potentially greater heterogeneity in terms of exposure and outcome [15]. However, they are typically simplistic in their depiction of environmental characteristics pertaining to behaviours of interest, and this simplification is often driven by limitations in data availability. For example, studies of food outlet location typically categorise outlets as simply healthy or unhealthy. Unhealthy outlets include small convenience stores and fast-food restaurants, whereas healthy outlets include greengrocers, farmers markets, and supermarkets [16]. Supermarkets provide access to a range of healthy foods, including fresh fruit and vegetables, and are seen as key indicators of a healthy food environment [16,17]. However, although supermarkets offer healthy food items, they are clearly also sources of unhealthy foods. This is important because it is the nature of the foods bought (and then consumed) that impacts health rather than shops’ geographical proximity to people’s homes. Consideration of how people use supermarkets and the types of food they buy are typically only studied in smaller-scale studies focused on single stores or neighbourhoods. For example, Cannuscio [11] conducted interviews and food environment audits in Philadelphia to explore residents’ shopping behaviours in relation to healthy foods. They found that store and food choices were related to a variety of social needs, not simply proximity and availability; however, the small sample size may have limited generalisability to the population level.

The rise of ‘big data’ datasets characterised by large size, complex nature, and the ability to link with other datasets [18], provides the potential to utilize better information on behaviour relevant to population-level health studies. In obesity research, the use of big data has been limited to date but has the potential to provide important contributions [19]. One example of such data that hold potential in this field is store-level sales data. These data are routinely collected by individual store chains to manage stock but are increasingly made available to researchers. Sales data allows for consideration of how stores are used, not just where they are located. Research is now being conducted using such data, particularly at relatively small geographic scales, assessing changes in purchasing patterns as interventions are trialled [20,21]. For example, Fuller et al. [21] used data on sales at a newly opened grocery store to explore what local residents purchased, finding that those who had previously lived in a food desert neighbourhood spent more on fruit in the new store than those who had previously had better access to fruit.

Using big data, we previously explored the association between supermarket sales and childhood overweight and obesity in England. We found that increasing sales of unhealthy foods were associated with higher rates of obesity and overweight among 4–5-year-olds and 10–11-year-olds [22]. In this paper, we investigate whether similar associations exist for adults and build on previous work by considering the causal pathway between food sales and obesity and whether the consumption of healthy foods (fruits and vegetables) mediates any relationship between food sales and obesity.

Using food sales data from a major supermarket chain in England and anthropometry and diet data of a large population-level sample (Active People Survey) we aim to answer the following questions:Are unhealthy food sales related to weight status and BMI?Are unhealthy food sales related to consumption of healthy foods?Does consumption of healthy foods mediate the relationship between sales of unhealthy foods and BMI?

Based on our findings, we will more broadly assess the utility of supermarket sales data in area-level public health research.

## 2. Materials and Methods

In this study, we used two main datasets. Data on store-level food sales were obtained for a national United Kingdom supermarket chain (which cannot be named, but includes shoppers across the social spectrum) through a customer data science company. This provided the food environment measure in our analysis. Food sales were aggregated to areas (English local authorities) to enable them to be joined to a separate dataset containing self-reported anthropometry and fruit and vegetable consumption (our primary outcomes), along with several potential covariates for adults across England. The methodology used is detailed below.

### 2.1. Study Population and Anthropometric Measurements

The study population comprises participants in the Active People Survey (APS). APS was conducted annually from 2005 to 2016 by Sport England (a UK government-funded body aiming to increase community-level sports, with an increased focus on physical activity and health) to monitor sports participation in English adults (age 16 years and older). However, it also included questions on weight, height, and fruit and vegetable consumption. APS was constructed to be representative of the English adult population and is available for research use through the UK Data Service [23]. Full details of the APS methodology are available elsewhere [24], but to summarise, random digit dialling was used to select landline-only numbers from a database of all exchange codes allocated for residential use in the UK. A respondent (aged 16 or over) was selected randomly within each household. Selected respondents were asked to report participation in a wide range of sport and physical activities, along with their age, sex, height, weight, and consumption of fruit and vegetables. The APS aimed to conduct a minimum of 500 interviews annually in each of the 326 local authorities of England (local government zones with populations ranging from 35,000 to 1.1 million and areas ranging from 12 km^2^ to 5014 km^2^). We used the data at individual level from APS 8, which were collected between 15 October 2013 and 14 October 2014.

Self-reported height and weight were used to calculate body mass index (BMI: weight (kg)/height (m)^2^), and participants were classified as overweight or obese based on standard cut points (BMI 25 and 30, respectively [25]). Questions on the consumption of fruit and vegetables were included in APS for the first time in APS 8. They asked participants:


*“How many portions of fruit did you eat yesterday? Please include all fruit, including fresh, frozen dried or tinned fruit, stewed fruit or fruit juices and smoothies*
*.”*
[24]

and:


*“How many portions of vegetables did you eat yesterday? Please include fresh, frozen, raw or tinned vegetables, but do not include any potatoes you ate.”*
[24]

APS had previously trialled different versions of these questions, comparing the impact of asking about ‘usual’ consumption and consumption ‘yesterday’. Interviews with participants after being asked these questions suggested that ‘yesterday’ was more easily understood and yielded more precise answers [24].

As well as our primary outcomes (BMI, weight status, and fruit and vegetable consumption), data on key covariates were also available in APS. Age and gender were self-reported in the survey, and as a measure of overall physical activity, minutes per week of moderate to vigorous physical activity (MVPA) were derived from reported time spent in all sporting and non-sporting physical activities. Based on weekly MVPA, participants were categorised as inactive (<30 min per week), fairly active (30–149 min per week), or active (≥150 min per week).

Although the outcome variables and covariates measured in APS 8 were available at the individual level, the only information on where APS participants lived was their local authority of residence.

### 2.2. Food Sales Data

Data on food sales, our primary exposure measure, were obtained from a large supermarket chain comprising 2482 stores across Great Britain. The data were extracted on food purchased from nine food categories during a 52-week period covering mid-August 2012 to mid-August 2013, based on a 10% sample of the retailer’s eighteen million loyalty-card holders. The categories were: fresh and frozen fruit and vegetables, cakes, biscuits, savoury pies, savoury snacks, and sweetened drinks. We chose these food categories, as they can be classified relatively unambiguously as either “healthy” or “unhealthy”. Fruit and vegetables are typically taken to be synonymous with a healthy diet, whereas foods such as cakes, biscuits, savoury pies and snacks, and sweetened drinks are considered unhealthy due to high fat and/or sugar content, and consumption should be limited [26]. In total, our data included sales of 1.02 billion healthy stock keeping units (SKU) and 601 million unhealthy SKU.

A geographic analysis was undertaken to aggregate the store-based food sales data to local authorities, the only measure of geography included in the APS. In order to link the two datasets, the postcodes of supermarket stores were geocoded and allocated to the local authority area in which they were situated in a geographical information system (ArcGIS 10.2 [27]). As people may travel beyond their home local authority to shop, we summed sales from all stores within 10 km of each local authority. This distance is equivalent to the average distance people reported traveling to shop in the UK [28], although we conducted sensitivity analysis with buffer distances of 0 km, 5 km, and 15 km around local authorities, showing very similar results (findings not presented). Sales from all stores within each local authority and its surrounding area were summed.

Because the absolute volume of sales varies between stores and local authorities, we used the composite measure of food sales healthiness developed for our study on childhood obesity [22]. This measure is the sales of unhealthy foods as a percentage of total sales for the nine food categories (hereafter, unhealthy foods sales percentage (UFSP)) for each local authority.

### 2.3. Data Linkage and Statistical Analysis

Aggregated food sales data as described above were linked to the APS data via local authority. Each APS participant was assigned a UFSP based on the local authority they lived in. We then used a regression-based approach to explore associations between area-level UFSP and individual-level weight status and fruit and vegetable consumption. As our data are hierarchical, with individuals nested within local authorities, and basic regression models assume that observations are independent (i.e., not nested), we used robust standard errors that relaxed these assumptions, allowing for intragroup (within local authority) correlation.

We used linear regression models to explore the association between UFSP and log-transformed BMI and applied binary logistic models for UFSP and the odds of being overweight or obese. Because fruit and vegetable consumption represent count data (number of portions per day), negative binomial regression models were used to test the association between UFSP and fruit and vegetable consumption. For the fruit and vegetable outcomes, separate models were run for fruit and vegetables (i.e., one model with fruit consumption as the outcome and another for vegetable consumption), as well as another model with combined fruit and vegetable consumption. All regression models were adjusted for sex, age group, individual-level physical activity, area-level index of multiple deprivation (IMD), and population ethnicity (% non-white) from the UK 2011 Census data [29]. These correlates have been associated with increased risk of obesity [30]. For ease of interpretation, odds ratios were plotted from the logistic regression models, and the linear and negative binomial models were used to predict BMI and fruit and vegetable consumption for quintiles of UFSP. To explore the mediation of the relationship between UFSP and BMI by fruit and vegetable consumption, we considered three regression models and estimated four parameters: the independent variable (IV) UFSP, as well as the dependent variable (DV) BMI by the mediator variable (MV) fruit and vegetable consumption.

The first model was the dependent variable regressed on the independent variable (here, BMI on UFSP). This provides the total effect of UFSP on BMI. The second model was the mediator regressed on the independent variable (here, fruit and vegetable consumption on UFSP). The third model was the dependent variable regressed on both the mediator and independent variable (BMI on both fruit and vegetable consumption and UFSP). The total effect is equal to the direct effect plus the indirect effect. Estimation of these parameters in regression models allows us to estimate the size of the effect of UFSP on BMI, as well as the proportion of that effect explained by fruit and vegetable consumption. The algorithm used to undertake the mediation analysis utilised bootstrapped confidence intervals as opposed to those based on robust standard errors [31,32]. The fruit and vegetable was considered both separately and combined. All analyses were carried out in Stata version 11 [33].

## 3. Results

All 326 local authorities in England were represented in APS. However, the Isles of Scilly, an island local authority off the southwest coast of England, did not have a store and was removed from analyses, leaving a sample of 325 local authorities. APS 8 included data on 126,084 individuals from these 325 local authorities. Of these 10,193 were excluded, as they did not report either height or weight, and therefore, BMI could not be calculated. A further 2876 did not answer questions on fruit and vegetable consumption, and 1729 did not report age and were excluded from our analyses. This left a sample of 111,287 individuals (88%), an average of 342 per local authority.

Sales data were available for all the chain’s 2,482 stores in Great Britain, of which 2063 were within 10 km of an English local authority and were therefore included in our analysis. This included some stores located in Wales (*n* = 12) and Scotland (*n* = 1) but within 10 km of English local authorities on the borders of Wales and Scotland and which could therefore be used by residents of English local authorities. Summary statistics describing the individuals and local authorities included in these analyses are presented in Table 1. In terms of individuals, 58% were female, over half (57%) were aged 55 and older, 30% reported being inactive (less than 30 min MVPA per week), and 52% were overweight or obese (17% obese). Nationally in England, 26% of the adult population is obese, and 26% consume at least five portions of fruit and vegetables per day [34]. Comparison with national figures suggests that our sample is less obese, consumes more fruit and vegetables, is older, and comprises a larger proportion of female participants.

Table 2 shows the results of the logistic regression models predicting the odds of being overweight or obese based on UFSP, and Figure 1 graphically displays these associations. The odds of being overweight or obese increased with increasing UFSP in the local authority of residence. The odds of being overweight or obese were 1.18 (95% CI, 1.10–1.26) and 1.18 (95% CI: 1.09–1.27), respectively, for those in the highest quintile of UFSP compared to those in the lowest quintile. Although there was a slight drop in odds of overweight and obesity in the fifth quintile compared to the fourth quintile, linear tests for trend were statistically significant for both outcomes (*p* < 0.05), and using the continuous version of the UFSP variable produced similar results. Estimates for the odds ratios for a unit increase in UFSP were 1.01 (95% CI: 1.005–1.014) for overweight and obese and 1.009 (95% CI: 1.005–1.013) for obese (both *p* < 0.001). The odds ratios observed for increasing quintiles of UFSP were of a similar magnitude to those for quintiles of deprivation; living in the most deprived local authorities was associated with a 21% increase in the odds of being overweight (95% CI: 14–28%). The outcome BMI (Figure 2) shows the same pattern, with increased BMI along with increasing local authority UFSP, although the effect appears small; the beta coefficient for the continuous UFSP was 0.001 (95% CI: 0.0006–0.0014, *p* < 0.001), with a predicted difference of 0.21 BMI points between the 10th and 90th centiles.

Figure 3 shows the association between UFSP and the reported consumption of fruit and vegetables. For both, mean consumption decreased with increasing values of UFSP; in areas where unhealthy foods make up a higher proportion of total sales, APS participants reported eating fewer portions of fruit and vegetables. However, although statistically significant associations exist for both fruit and vegetables, the difference between the 10th and 90th centiles of UFSP was less than 0.1 of a portion per day.

Table 3 describes the results of mediation analysis exploring whether the relationship between UFSP and BMI is mediated by fruit and vegetable consumption. Fruit consumption showed a pattern of mediation whereby higher UFSP was associated with lower consumption of fruit; higher consumption of fruit was in turn was associated with higher BMI. The pattern was different for vegetable consumption in that higher vegetable consumption was associated with lower BMI. Although the effects were statistically significant for fruit and vegetables separately, in each case, the indirect effect (the portion of the overall effect of UFSP on BMI via consumption) was very small. The percentage of the total effect moderated was less than 1.5% for all three moderators.

## 4. Discussion

Using a novel combination of a large dataset on food sales from a national retailer and a population-level dataset on weight status and food consumption, our results show a clear association between supermarket sales of unhealthy foods as a percentage of overall sales (UFSP) and BMI and the odds of overweight and obesity in adults. An association was also seen between UFSP and the consumption of healthy foods, which mediated the overall relationship seen between sales and adiposity, although only to a very small extent.

The use of supermarkets as proxy for access to healthy food is common [16,35,36,37], as their presence increases the local availability of healthy foods, especially fruit and vegetables [38]. Large supermarket chains, such as the one included in this study, do indeed offer good access to healthy foods, such as fruit and vegetables, compared with other food outlets (e.g., discount stores) [39], but they also provide unhealthy foods, increased access to which may amplify existing poor eating habits [40]. Our results show that the sales of unhealthy foods relative to total sales at supermarkets is associated with higher rates of obesity in the local population, supporting the need to consider store use and location. These results echo the findings of a review by Dicken and Batterham [5], as the unhealthy foods in our analyses correspond to ultra-processed food consumption, which was significantly associated with obesity and poor health outcomes. Interventions within stores to promote the purchasing of healthier foods have been trialled [41,42], and such strategies should be considered alongside the addition of new stores in underserved areas.

In addition to exploring the associations between supermarket sales and weight status, these analyses allow us to consider the utility of large-scale sales data in food environment studies. There are some clear strengths in the food sales data. They offer large sample sizes, covering a national scale, which potentially provides greater heterogeneity in both exposures and outcomes than smaller-scale data. Although sales were from one store chain only in our study, that chain has the largest market share in the UK, with an estimated 18 million loyalty-card holders (around a third of the UK adult population). The detail contained within the data allows researchers to explore patterns in sales of specific food types or to aggregate to categories of food.

A consistent method of data collection (e.g., the same information about the same categories of food) and the temporal scale at which data are available allow an exploration of how sales patterns and potential associations with health outcomes vary over time. We aggregated data over a year to approximate average sales, but data are available at the day-level, which means they can be used to derive temporal measures of food purchases. In our case, one year of data balanced out any seasonal fluctuations in sales [43]. The data are available in readily readable formats and include consistent categorisation in terms of food types. All sales are linked to the store in which they were made, for which full address and postcode is available, making geolocation of stores precise. Further, the fine spatial scale of the data means that they can be aggregated as necessary to enable links to be made with other datasets using different geographies.

Despite a range of advantages, there are limitations in the use of such data. Beyond some market segmentation categories [44], the data provide no information about the customers buying the food. To use these supermarket sales data for health research, they must be linked to another dataset that provides information on health behaviours or outcomes, as this information is not available for shoppers in each store. Store location is therefore the only means by which sales data can be linked to other datasets. In a study such as this, exposure and outcome measures are thus drawn from different samples, and analyses are potentially subject to problems such as the ecological fallacy, whereby associations seen between groups do not necessarily apply to individuals [45]. The lack of demographic information in the sales data means that we cannot tell how representative these shoppers are of all shoppers at this store chain (as the data are based on a sample of loyalty card holders), nor the wider population. It may also be the case that purchases made with a loyalty card are not representative of all purchases, as loyalty card use is less prevalent in smaller store formats.

When using store location data to describe local food environments, researchers assume a causal pathway on which residents use their local store to purchase food that is then consumed. The healthiness and quantity of this food then contributes to weight status. Use of sales data is a step along this potential causal pathway (e.g., it allows for consideration of how stores are used, not just where they are located); however, it is still not a measure of the key behaviour that causes weight gain. Foods must be consumed, and these data do not provide information on what foods are eaten. As the food sales data and weight status are drawn from different samples, many in the APS sample may not use this supermarket chain, not all the food sold at this store chain will be eaten, and people will also consume food bought in other shops and restaurants.

These sales data contain a huge amount of detail in terms of individual food items, allowing researchers to distinguish, for example, between assorted brands of specific items. Food items are necessarily grouped into categories, but these are not all unambiguously healthy or unhealthy; bakery foods, for example, include doughnuts and wholemeal bread. For these analyses, we selected only food categories that were clearly healthy or unhealthy; this meant that only a small subset of the food items sold by each store were considered. We used a relative measure of sales healthiness rather than absolute sales volumes so that we could aggregate data across local authorities. This meant that we were not able to distinguish between sales at different high- and low-volume stores, which may have different sales patterns.

These analyses used sales data to approximate average sales of healthy food within an area and thus describe the food environment. Although the sales data we used were from a nationwide chain with high market share, they do not represent all purchases. It is widely known that food purchases and consumption are socially patterned [46,47]; therefore, area-level sales may simply reflect area-level socioeconomic status. We included a measure of area deprivation in our models (IMD), and UFSP was independently significantly associated with the odds of obesity, with an effect in the same direction and of a similar order of magnitude as that seen for IMD. This suggests that these data provide important additional explanatory power to this form of analysis. The fact that the mediation analysis showed only a very small proportion of the effect of UFSP on BMI was via the anticipated pathway could be due to data limitations, as measures of sales and consumption were drawn from different samples.

There were also some limitations with the APS dataset. Although it provided a large, nationwide sample with comprehensive measures of physical activity, weight status, and fruit and vegetable consumption, these variables were all measured through self-report. Weight status based on self-reported information may vary in its accuracy by sex, age, and weight status [48,49]. Although self-report measures are deemed appropriate for the measurement of population-level adiposity [48], underreporting of weight by obese participants has been noted [49]. This may have attenuated any associations seen, possibly contributing to the small effect sizes observed. Similarly, the measures of fruit and vegetable consumption were also self-reported, and the exact questions used in APS 8 have not been validated. However, similar questions have been found to show substantial test–retest reliability and good validity, in particular in terms of ranking individuals in terms of fruit and vegetable consumption [50]. The APS sample was drawn from households with landlines (82% of all UK households [51]), which could slightly reduce population representativeness. Comparison with national statistics suggests that the APS sample has a lower prevalence of obesity and higher fruit and vegetable consumption than the wider population. Individual-level measures of socioeconomic status were not available in the APS 8 data that we had, so we had to use area-level demographic variables in our models, potentially reducing our power to detect associations.

The use of big data in public health should aim to better clarify associations, an important step in discovering and unpicking causal relationships, allowing interventions to be planned to improve health [18]. Sales data have the potential to provide more insight into associations between food environments and health than store location data and can suggest where future work may be best targeted. For example, such data could be used to target dietary interventions at the store level and monitor changes in shopping habits over time. However, Attree [52] argues that reliance on consumer behaviour does nothing to help people living on low incomes who have few options other than to choose cheap, unhealthy foods and urges for change in health policies to address this.

Currently, policies to tackle obesity and the associated health problems are mainly focused on supporting individuals and reformulating foods to make them ‘healthier’ in terms of salt, sugar, and fat reduction [53,54]. Although laudable, such policies do not address what research is showing to be driving obesity: purchase and consumption of unhealthy, highly processed food.

## 5. Conclusions

Our findings show a clear association between the sales of unhealthy foods within a supermarket chain and odds of overweight and obesity among adults in the locality. However, this association does not seem to be strongly mediated by the logical route of consumption of healthy foods. These results suggest that purchasing behaviour within food outlets, as well as outlet location, should be considered in future work on food environments. Development of future health policy to tackle obesity should consider using ‘big data’ combined with other research methods to address the increasing consumption of unhealthy and highly processed foods.

## Figures and Tables

**Figure 1 nutrients-14-01745-f001:**
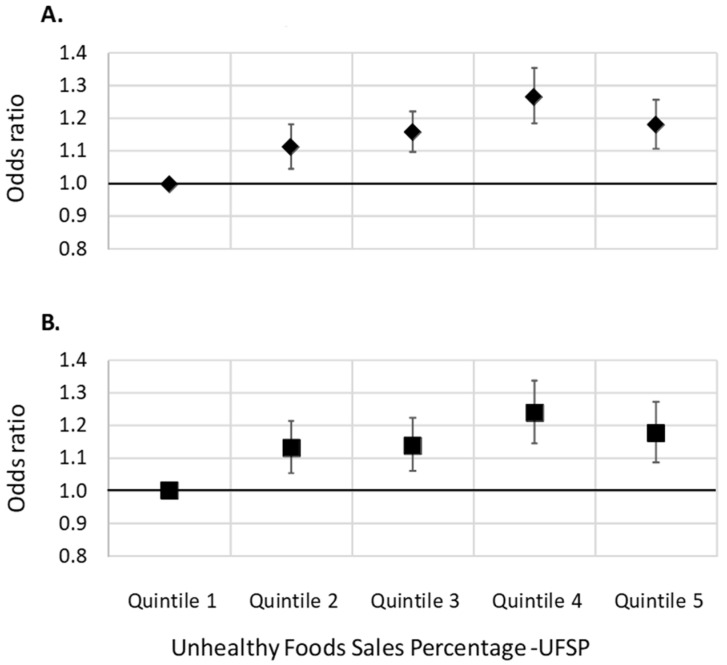
Associations between unhealthy food sales percentage and the odds of being (**A**) overweight (including obese) or (**B**) obese. Odds ratios from logistic regression models (ref = Q 1). Models adjusted for sex, age group, and individual-level physical activity, as well as local-authority-level deprivation and ethnic mix.

**Figure 2 nutrients-14-01745-f002:**
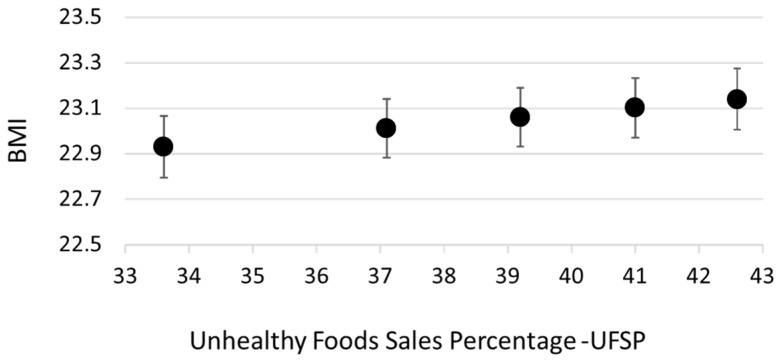
Association between unhealthy food sales percentage and BMI. Figure shows predicted BMI at Q midpoints (10th, 30th, 50th, 70th, and 90th centiles) based on a linear regression model. Model adjusted for sex, age group, and individual-level physical activity, as well as local-authority-level deprivation and ethnic mix.

**Figure 3 nutrients-14-01745-f003:**
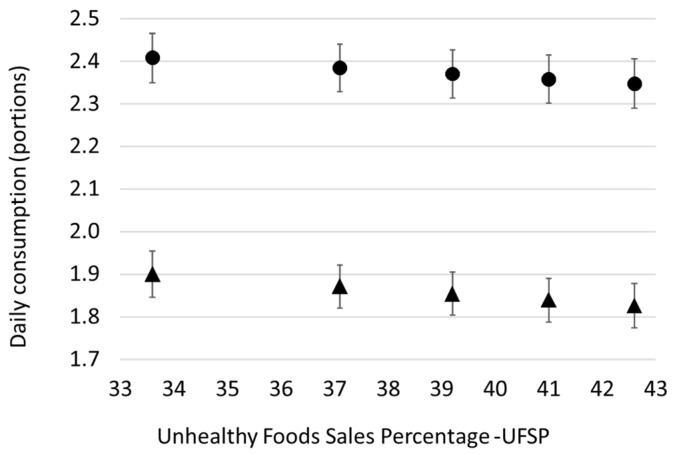
Associations between unhealthy food sales percentage and the self-reported consumption of fruit (l) and vegetables (p). Figure shows predicted consumption at Q midpoints (10th, 30th, 50th, 70th, and 90th centiles) based on negative binomial regression models. Models adjusted for sex, age group, and individual-level physical activity, as well as local-authority-level deprivation and ethnic mix.

**Table 1 nutrients-14-01745-t001:** Descriptive statistics of individual participants in the Active People Survey and English local authorities.

	*n* (%) or Median; IQR (Unless Otherwise Stated)
Individuals (from APS)	
Total *n*	111,287
Gender	
Female	64,893 (58.3%)
Male	46,394 (41.7%)
Age group	
16–24	5872 (5.3%)
25–34	7792 (7%)
35–44	14,258 (12.8%)
45–54	19,515 (17.5%)
55–64	21,586 (19.4%)
65–74	22,989 (20.7%)
75+	19,275 (17.3%)
Physical activity	
Inactive (<30 min MVPA a week)	33,727 (30.3%)
Fairly active (30–149 min MVPA a week)	16,849 (15.1%)
Active (≥150 min MVPA a week)	60,711 (54.6%)
Physical activity	
Not overweight/obese	53,253 (47.9%)
Overweight	39,311 (35.3%)
Obese	18,723 (16.8%)
Fruit and vegetable consumption	
Vegetables/day	2; 1–3
Fruit/day	3; 1–4
Total Fruit & Vegetables/day	5; 3–7
Eat ≥5 portions per day	63,982 (57.5%)
Local Authorities	
Total *n*	325
UFSP for all stores within the local authority + 10 km (Mean (SD))	39.3 (4.8)
Total population	125,746; 95,262–202,228
Percentage non-white	5.1%; 2.6–12.6%
Average deprivation score (Mean (SD))	19.5 (8)
Components of deprivation	
Percentage of the population with bad or very bad general health	5.1%; 4.2–6.1%
Percentage of total population age 25+ with highest qualification < Level 2	37.1%; 32.2–41.6%
Percentage of households with 1 or more people per room	1.1%; 0.8–1.7%
Percentage unemployed ‘Economically active: unemployed’/All usual residents 16–74	3.7%; 3–4.8%

Note: food sales represent units sold in 2012–2013, weight status, and physical activity for APS8 collected in 2013–2014. Demographic details for local authorities are from the 2011 UK Census.

**Table 2 nutrients-14-01745-t002:** Full results of multivariate logistic regression model predicting odds of overweight and obesity by UFS.

		Overweight & Obese				Obese			
		OR	95% CI		*p*	OR	95% CI		*p*
Gender	Female	1				1			
	Male	1.757	1.711	1.804	<0.001	1.148	1.112	1.184	<0.001
Age group	Age 16–24	1				1			
	Age 25–34	2.502	2.322	2.696	<0.001	2.255	1.983	2.564	<0.001
	Age 35–44	3.213	2.993	3.450	<0.001	2.472	2.193	2.786	<0.001
	Age 45–54	4.165	3.887	4.462	<0.001	3.265	2.905	3.670	<0.001
	Age 55–64	4.470	4.172	4.790	<0.001	3.278	2.928	3.671	<0.001
	Age 65–74	4.288	3.996	4.602	<0.001	2.891	2.572	3.249	<0.001
	Age 75+	3.001	2.797	3.220	<0.001	1.701	1.515	1.909	<0.001
Physical activity	PA <30 min/week	1				1			
	PA 30–89 min	0.894	0.854	0.936	<0.001	0.788	0.742	0.837	<0.001
	PA 90–149 min	0.851	0.810	0.893	<0.001	0.696	0.657	0.738	<0.001
	PA 150 min+	0.647	0.628	0.666	<0.001	0.475	0.458	0.492	<0.001
Local Authority % population non-white ethnicity	Q1 (lowest %)	1				1			
	Q2	0.989	0.942	1.039	0.669	1.050	0.982	1.123	0.150
	Q3	0.960	0.915	1.008	0.101	1.011	0.943	1.083	0.766
	Q4	0.985	0.929	1.044	0.608	1.002	0.922	1.089	0.962
	Q5 (highest %)	0.929	0.872	0.991	0.025	1.006	0.934	1.083	0.875
Local Authority average IMD score	Q1 (least deprived)	1				1			
	Q2	1.083	1.031	1.138	0.002	1.127	1.056	1.203	<0.001
	Q3	1.120	1.060	1.183	<0.001	1.164	1.080	1.254	<0.001
	Q4	1.196	1.118	1.279	<0.001	1.229	1.131	1.334	<0.001
	Q5 (most deprived)	1.209	1.139	1.284	<0.001	1.270	1.177	1.371	<0.001
Unhealthy food sales percentage	Q1 (least unhealthy)	1				1			
	Q2	1.112	1.045	1.183	0.001	1.131	1.054	1.214	0.001
	Q3	1.157	1.097	1.221	<0.001	1.140	1.062	1.224	<0.001
	Q4	1.268	1.187	1.354	<0.001	1.239	1.146	1.340	<0.001
	Q5 (most unhealthy)	1.181	1.109	1.258	<0.001	1.177	1.088	1.274	<0.001

Reference categories for the outcomes are normal weight only for the overweight and obese models and normal and overweight for the obese models.

**Table 3 nutrients-14-01745-t003:** Results of mediation analysis exploring fruit and vegetable consumption as mediators in the relationship between local authority unhealthy food sales percentage and BMI.

	β Coef.	SE	95% CI	
Mediator: Fruit and Vegetable consumption ^a^				
(coefficient for IV on MV)	−0.01489	0.00197	−0.01876	−0.01103
(coefficient for MV on DV)	0.00027	0.00020	−0.00011	0.00066
Total effect	0.00101	0.00013	0.00075	0.00126
Direct effect	0.00101	0.00013	0.00076	0.00126
Indirect effect *	0.00000	0.00000306	−0.00001	0.00000
Mediator: Fruit consumption ^b^				
(coefficient for IV on MV)	−0.00642	0.00135	−0.00907	−0.00377
(coefficient for MV on DV)	0.00187	0.00028	0.00132	0.00243
Total effect	0.00100	0.00013	0.00075	0.00125
Direct effect	0.00101	0.00013	0.00076	0.00126
Indirect effect *	−0.00001	0.00000323	−0.0000183	−0.00000568
Mediator: Vegetable consumption ^c^				
(coefficient for IV on MV)	−0.008106	0.001158	−0.01038	−0.00584
(coefficient for MV on DV)	−0.001782	0.000332	−0.00243	−0.00113
Total effect	0.001008	0.000129	0.00076	0.00126
Direct effect	0.000994	0.000129	0.00074	0.00125
Indirect effect *	0.000014	0.00000347	0.00000765	0.0000212

Percent mediated: ^a^ 0.4%, ^b^ 1.2%, ^c^ 1.4%; * indirect effect SE and 95% CI are all based on bootstrapping; IV = independent variable (unhealthy food sales percentage); MV = mediator variable (fruit and vegetable consumption, fruit consumption, or vegetable consumption); DV = dependent variable (BMI for all models).

## Data Availability

The sales data used in this study from the supermarket are not publicly available due to commercial restrictions. The Active People Survey data are available from Sport England, https://activepeople.sportengland.org/, accessed on 1 January 2022.

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
