# Peer review of "Food Sales and Adult Weight Status: Results of a Cross-Sectional Study in England"

_nutrients, 2022, doi:10.3390/nu14091745_

Round 1
Reviewer 1 Report
The authors, using big data from food sales and anthropometry and diet information from a large sample, tried to understand the relationship between unhealthy food sales and weight status/BMI, between unhealthy food sales and the consumption of healthy foods (fruit and vegetables), and if the consumption of healthy foods mediates the relationship between sales of unhealthy foods and BMI.
The authors have rightly emphasized the potential of big data applied to public health. The paper is interesting and well written. I have just a few suggestions to improve the clarity of the manuscript. In particular, I found Table 3 hard to read. It is unclear if the first five rows refer to fruit consumption as a mediator. It could be more useful two graphs (one for fruit consumption and one for vegetable consumption) similar to figure 1 (that could be deleted from the methods).
Some tips:
Line 66: “Big Data;” could be ”Big Data,”.
Author Response
Table 3 shows the results of all three of the mediation analyses we performed considering fruit and vegetable consumption (separately and combined) as mediators in the relationship between UFSP and BMI. The first five rows are the results for combined fruit & vegetable consumption as the mediator. We have now provided a heading for the first five lines in the table. We feel it is important to include full details of the model results in the paper (including measures of the variability of model estimates; standard error and confidence intervals). This volume of information would become very confused on a diagram such as that used in figure 1, so we feel they are more clearly displayed in a table. We felt it important to run all three mediation models as fruit & vegetables are often considered together in terms of health policy (e.g. the recommendation to eat 5 portions per day), but they potentially show different associations with behaviour and health outcomes (as seen here). Interpretation of the numbers in table 3 is given in the text.
Thank you - Line 66 has been changed to reflect what the reviewer said 'Big Data'.
Reviewer 2 Report
The authors explored the food sales and adult weight status in England via a cross-sectional study. It is a wonderful research to understand the associations among food sales, adult weight status and potential factors, which is helpful to provide evidences for the formation of health policy.
I have a suggestion that more detail are needed in the Introduction and Discussion to clarify the importance of the research to health policy.
Author Response
Thank you to Reviewer 2 for the suggestion of adding healthy policy to the manuscript. We have added a sentence to the abstract and conclusion about health policy and then added a substantial paragraph to the paper:
Abstract- Future health policies should consider using ‘Big Data’ combined with other research to address the increasing consumption of unhealthy and highly processed foods.
Conclusion - Development of future health policy to tackle obesity should consider using ‘Big Data’ combined with other research to address the increasing consumption of unhealthy and highly processed foods.
Discussion - For example, the data could be used to target dietary interventions at store level and monitor changes in shopping habits over time. However, Attree 52 argues that reliance on consumer behaviour does nothing to help people on low incomes who have little option but choose cheap, unhealthy foods and urges for change in health policies to address this.
Currently, policies to tackle obesity and the associated health problems are mainly focused on supporting individuals and reformulating foods to make them ‘healthier’ in terms of salt, sugar, and fat reduction 53,54. While laudable, the policies do not address what research is showing to be driving obesity – purchase and consumption of unhealthy highly processed food.